# Predicting stress in first-year college students using sleep data from wearable devices

**Laura S. P. Bloomfield**[1,2,3]\*, **Mikaela I. Fudolig**[2,3], **Julia Kim**[3], **Jordan Llorin**[3], **Juniper L. Lovato**[3], **Ellen W. McGinnis**[4,5], **Ryan S. McGinnis**[5,6], **Matt Price**[3,7], **Taylor H. Ricketts**[1,8], **Peter Sheridan Dodds**[3,9], **Kathryn Stanton**[3], **Christopher M. Danforth**[1,2,3]\*

**1** Gund Institute for Environment, University of Vermont, Burlington, Vermont, United States of America, **2** Department of Mathematics & Statistics, University of Vermont, Burlington, Vermont, United States of America, **3** Vermont Complex Systems Center, University of Vermont, Burlington, Vermont, United States of America, **4** Department of Social Science and Health Policy, Wake Forest University School of Medicine, Winston-Salem, North Carolina, United States of America, **5** Center for Remote Patient and Participant Monitoring, Wake Forest University School of Medicine, Winston-Salem, North Carolina, United States of America, **6** Department of Biomedical Engineering, Wake Forest University School of Medicine, Winston-Salem, North Carolina, United States of America, **7** Department of Psychological Science, University of Vermont, Burlington, Vermont, United States of America, **8** Rubenstein School of Environment and Natural Resources, University of Vermont, Burlington, Vermont, United States of America, **9** Department of Computer Science, University of Vermont, Burlington, Vermont, United States of America

\* lbloomfi@uvm.edu; chris.danforth@uvm.edu

**Data Availability Statement:** We have provided data as supplementary information and with which results are replicable. We have removed all data

## Abstract

Consumer wearables have been successful at measuring sleep and may be useful in predicting changes in mental health measures such as stress. A key challenge remains in quantifying the relationship between sleep measures associated with physiologic stress and a user's experience of stress. Students from a public university enrolled in the Lived Experiences Measured Using Rings Study (LEMURS) provided continuous biometric data and answered weekly surveys during their first semester of college between October-December 2022. We analyzed weekly associations between estimated sleep measures and perceived stress for participants (N = 525). Through mixed-effects regression models, we identified consistent associations between perceived stress scores and average nightly total sleep time (TST), resting heart rate (RHR), heart rate variability (HRV), and respiratory rate (ARR). These effects persisted after controlling for gender and week of the semester. Specifically, for every additional hour of TST, the odds of experiencing moderate-to-high stress decreased by 0.617 or by 38.3% ($p$<0.01). For each 1 beat per minute increase in RHR, the odds of experiencing moderate-to-high stress increased by 1.036 or by 3.6% ($p$<0.01). For each 1 millisecond increase in HRV, the odds of experiencing moderate-to-high stress decreased by 0.988 or by 1.2% ($p$<0.05). For each additional breath per minute increase in ARR, the odds of experiencing moderate-to-high stress increased by 1.230 or by 23.0% ($p$<0.01). Consistent with previous research, participants who did not identify as male (i.e., female, nonbinary, and transgender participants) had significantly higher self-reported stress throughout the study. The week of the semester was also a significant predictor of stress. Sleep data from wearable devices may help us understand and to better predict stress, a strong signal of the ongoing mental health epidemic among college students.

that could breach compliance with our ethics board to maintain confidentiality of participants.

**Funding:** LSP Bloomfield was supported by the Gund Fellowship and partial salary from the Mass Mutual Insurance Wellness Initiative (Grant # FP2860). JK, JL, and KS salaries are funded by the Mass Mutual Insurance Wellness Initiative (Grant # FP2860). EWM was supported by the U.S. National Institute of Health under Grant MH123031. This project has been funded as part of the Lived Experience Measured Using Rings Study (LEMURS) through the Mass Mutual Insurance Wellness Initiative (Grant # FP2860). The funders had no role in study design, data collection and analysis, decision to publish, or preparation of the manuscript.

## Author summary

College students are at high-risk for mental-health related morbidity and mortality. Determining which objective sleep measures are associated with stress provides an opportunity to identify who is at risk and intervene in real time. We enrolled a large cohort of first-year college students to assess the relationship between self-reported stress and sleep estimates from wearable devices. We show that nightly averages of total sleep time, resting heart rate, heart rate variability, and respiratory rate are significantly associated with perceived stress during the first semester of college, controlling for gender and week of the semester. These findings establish the potential importance of physiologic estimates from wearable devices to predict stress in first-year college students.

## Introduction

Chronic stress has been linked to behavioral changes and adverse health outcomes [1–4]. Recent work has sought to connect measures of sleep with markers of mental health, including stress [5,6]. Specifically, elevated stress affects sleep quality and quantity [7–9]. The transition to college is considered a stressful life transition for many [10]. Coincident with the onset of common mental health disorders, such as anxiety and depression, college is a period marked by insufficient sleep and irregular sleep patterns [11]. A recent multi-university study found that more than 60% of college students met the criteria for poor sleep and mental health symptoms were associated with decreased sleep quality [12]. There is a growing literature showing that sleep is an important predictor for success in college [13,14], daily measures of mental health [15,16], and health outcomes later in life [17,18].

However, the relationship between stress and sleep responses does not follow a uniform dose-response due to between and within individual differences over time [19]. The chronicity of stressors and individual coping behaviors also influence the impact of stress on sleep [20–22]. Additionally, demographic, and psychological differences, including mental health diagnosis and exposure to adverse life events in childhood influence the magnitude of the impact of stress on sleep measures, particularly during transitional periods [23–25]. Previous evidence for the relationship between stress and sleep often lacks information on previous psychological history, are carried out in laboratory settings, are based on self-reported sleep data or are small sample sizes [26,27]. Absent experimental protocols, repeated measures of sleep and stress across a large population are needed to observe sufficient variation within and between individuals to assess whether sleep measures can be used to infer changes in mental health measures.

The widespread use of consumer-grade wearables makes it possible to rigorously estimate physiological and sleep measures in large-scale studies [13,28–30]. These studies have provided insight into the mechanistic and temporal relationships between sleep and stress. Previous work using machine learning on wearable sleep and mental health data has shown that sleep abnormalities are associated with the probability of mental illness onset [29]. Specifically, stress has been linked to reduced total sleep time (TST), increased sleep onset latency (SOL), and lower sleep efficiency [21,22,31,32]. Physiological indices measured during sleep, such as lower average nightly heart rate variability (HRV) [33–35] increased resting heart rate (RHR) [27], and variation in average nightly respiratory rate (ARR) [36,37] have also been linked to stressors. Recent work using skin conductance and temperature from wearable sensor data, has shown accuracy in classifying college students as high or low stress [30].

The Oura ring is a multi-sensor wearable device that has been validated for accurate sleep measurement [38–40]. In a 2021 validation study comparing the Oura ring to polysomnography (PSG), the Oura ring was 94% accurate in detecting sleep and wake periods when using an accelerometer-based model and 96% accurate with the inclusion of circadian features [28]. The Oura ring uses photoplethysmography (PPG) [41], which measures arterial pulses, to derive RHR, HRV, and ARR [42]. Night-time measurements are shown to be the most consistent due to day-time factors that can influence the accuracy of these measurements and introduce noise [43].

In the present longitudinal study, we investigated whether sleep measures from biometric data taken continuously with an Oura ring could be used to predict subjective measures of stress (PSS) in first-year college students. We hypothesized that reduced TST, higher RHR, lower HRV, and greater variation in ARR would be associated with higher PSS. We selected this critical window for evaluating the relationship because it is a major life transition associated with heightened stress and during which sleep behaviors change for many young adults. Given the health implications of prolonged stress and sleep disruptions, utilizing continuous biometric data to predict periods of heightened stress in young adults offers the potential for rapid assessment and targeted, preventative interventions.

## Materials and methods

### Enrollment

All participants were enrolled during the fall semester of their first year of university. Participants were recruited during orientation, through student mailing lists, and in-person events. After expressing interest in the study, participants were asked to complete basic demographic questionnaires, which were used to screen for eligibility. Inclusion criteria included being a first-year student between the ages of 18–24 years, being full-time (enrolled in at least 12 credits), and owning a smartphone. Interested participants who did not meet these criteria were excluded and not enrolled. After eligibility screening, participants were invited to attend a lecture which provided in-depth information about the study. Participants were required to complete a comprehension assessment with completely correct answers before being able to provide written informed consent through REDCap, a HIPPA-compliant online application. This study protocol was reviewed and approved by the University of Vermont Institutional Review Board (Protocol #2126).

After enrollment, each participant was assigned a unique study identifier. Participants then attended an in-person event to complete sizing for their Oura ring. The Oura ring automatically collected body response data during sleep and daily activity. That data was uploaded to Oura Cloud via the Oura mobile app and was accessed using the Oura mobile app or Oura on the web. When participants were enrolled in Oura Teams, all identifying information was removed and each participant was manually coded with their unique study identifier. Oura Teams data was only accessed by those approved by the UVM IRB. Additionally, Oura uses administrative, organizational, technical, and physical safeguards to protect the personal data that was collected and processed.

After receiving their Oura ring, participants were asked to complete a baseline survey and fill out weekly surveys. There were six additional weekly surveys that were distributed to participants. Responses to questions from electronic questionnaires were stored in REDCap. All data collected by this study was stored on a secure virtual machine on a drive that had restricted access to those who are approved by the UVM IRB.

### Weekly surveys

**Stress measure.** The PSS-10 is a ten-item measurement tool that assesses the degree to which individuals perceive of situations in their lives as uncontrollable, unpredictable, and overloaded relative to their subjective coping abilities (e.g., how often could a person not cope with activities and events in their life) [44]. Items were rated on a 5-point Likert-type scale (0 = never to 4 = very often) (S1 Text). Six items were considered the negativity subscale and four items related to the positivity subscale. The PSS-10 has internal consistency, with α of 0.91, and has also been widely validated across cultures [45]. The PSS-10 is not a diagnostic instrument, and there are not any clinically established cut-offs. However, it has been used as a screening tool, and scores above 14 have been considered a moderate-to-high level of stress [46]. Consistent with previous studies, we converted this score to a binary outcome variable at the threshold of 14 for moderate-to-high stress.

**Sleep measures.** To quantify sleep, the Oura ring uses a combination of accelerometer data, heart rate, heart rate variability, and pulse wave amplitude variability with machine learning models to calculate sleep duration, including those for deep sleep (N3), light sleep (N1+N2), rapid-eye-movement sleep (REM), and proportion of time a user is in bed awake [28]. The Oura ring has a high association with PSG for measuring TST, SOL, and wake after sleep onset (WASO) [35].

For each night, we used information recorded by the Oura ring for the sleep period with the longest duration (i.e., naps were not included). The measures from the Oura ring that we used in our models are given in the Supporting Information (S1 Table). In addition to the raw values, we also computed mean-adjusted values for participants. For a given individual, the adjusted measurements are $x_{\text{dev},k} = x_k - \text{mean}(\{x_1, x_2, \ldots, x_n\})$ where $k = 1, 2, \ldots, n$ is the $k^{th}$ night measurement of measure $x$.

To compare the weekly survey measures to the daily Oura sleep data, we aggregated the daily sleep measurements at a weekly level. For every user, we considered weeks where there were at least three days where sleep data was recorded. For each week, we computed the following statistics for the values in that week: minimum, the $5^{th}$, $25^{th}$, $50^{th}$ (mean), $75^{th}$, $95^{th}$ percentiles, and the maximum. By using these summary statistics as inputs to our model, we input a reduced representation of the sleep measures distribution for each participant for each week. We performed the same aggregation procedure on the adjusted daily values to adjust for within-individual variation.

**Compliance.** We observed a high level of compliance from participants who were eligible and enrolled in the study (N = 603) (S1 Fig). We filtered out participants who had less than three surveys to maintain integrity of the repeated measures analysis. 92.4% (557/603) of participants who enrolled in the study, completed at least three surveys. From the raw Oura data, 94.5% of user-week combinations (3530 out of 3735) had three or more Oura data points per week which corresponds to 98.7% (595/603) of enrolled participants with any days of Oura data. When weekly surveys were matched to weeks of Oura data with at least three nights of sleep data, 94.3% (525/557) of participants with at least three surveys remained in the sample (S2 and S3 Tables).

### Statistical analysis

**Mixed-effects models.** We used longitudinal mixed-effects models, also known as multi-level models, hierarchical linear models, or random effects models, due to the nested structure of panel data where repeated measures were taken [47]. These models handle missing and unbalanced data more effectively than traditional regression methods and can account for within-individual and between-individual variability [48]. The fixed effects represent the

average relationship between time-invariant traits and stress across the entire population [49]. We explored multiple model forms, but our final model was a mixed-effects linear regression model with random and fixed effects for week of the semester to account for differences in stress associated with the structure of the school year. The model included gender, week, and sleep measures as fixed effects which were not highly correlated (S4 Table). Participant identifier and week were included as random effects. This approach allowed us to control for within-subject variability across the weeks of the study, providing an estimate of the effect of these predictors on PSS. We used these models to predict PSS as a continuous and binary outcome measure. We also calculated weekly fluctuations in PSS with two additional outcome measures: 1) Change in PSS from previous measure (Δ PSS) and 2) Deviation of PSS from each participant mean (σ PSS). The equation of our model can be found in the Supplemental Information (S5 Table). We also explored mixed-effects models with nonlinear components to investigate Oura sleep measures as predictors of PSS (S2 Text).

## Results

Our data included weekly surveys and biometrics from a first-year college cohort (N = 525) (Table 1). 87% (525/603) of enrolled participants who had completed at least three weekly surveys had Oura ring data for at least three nights in the same week as their completed survey (*N* = 3,112). First-year college students slept, on average, 7.41 hours per night (*SD* = 0.83 hours). Participants had a mean individual variance of 5.15 hours of TST on weekdays during the study (*SD* = 1.49 hours) and an average variance of 2.72 hours (SD = 1.40 hours) in a single week (Table 1). Participants had an average RHR of 62.75 bpm (*SD* = 8.64 bpm). At the participant level, RHR varied by 18.90 bpm (*SD* = 8.38 bpm) over the course of the study and 7.89 bpm within a given week (*SD* = 5.62 bpm). Participants had an average HRV of 68.24 ms (*SD* = 33.09 ms) during the study. Participants had a mean individual variance in HRV of 57.95 ms (*SD* = 32.36 ms) over the course of the study and an average variance of 26.84 ms (*SD* = 20.64 ms) per week. Participants had a mean individual variance of ARR of 2.42 breaths per minute (*SD* = 1.03 breaths per minute) over the course of the study and an average variance of 1.02 breaths per minute (*SD* = 0.65 breaths per minute) per week (Table 1).

Weekly surveys revealed that the average Perceived Stress Score (PSS) for participants was 15.85 (*SD* = 7.33). 64.14% (1,996/3,112) of weekly survey responses indicated moderate-to-high stress (PSS> = 14). As far as fluctuations in stress, the average change in PSS between week was -0.48 (*SD* = 5.90) and the participant variance over the study was 12.07 (*SD* = 7.88). We also calculated the average PSS per participant and calculated their deviation from their average from week to week. On average, participants deviated from their average PSS by an average -1.15e-08 (*SD* = 4.11), or on average by 0 (Table 1). Participants who did not identify as male (i.e., female, nonbinary, and transgender students) had significantly higher PSS than male participants (S6 Table).

We calculated the intraclass correlation coefficients for TST, RHR, HRV, and ARR to assess the portion of variation attributable to within-subject variability. TST was the only predictor with a substantial portion of its variation attributable to within-subject variation (S7 Table). Conversely, more than 80% of the variation in RHR, HRV, and ARR was attributable to differences between participant measures rather than within participant measures. When added as predictors, these sleep measures did not have significant contributions to total variance in PSS attributable to between-individual differences (S7 Table). However, in univariate regression models, TST, RHR, HRV, and ARR were significant explanatory variables for PSS (S6 Table).

A relationship between PSS and sleep estimates (TST, RHR, HRV, and ARR) was observed across covariate-adjusted mixed-effects regression models for PSS as both a continuous and

**Table 1. Participant demographics, stress, and sleep characteristics of fall semester for 525 first-year college students from 3,112 Oura sleep measures.** Proportions, averages, and standard deviations of sleep measures with perceived stress scores (PSS).

| Demographic variables | % (N) | | |
|---|---|---|---|
| **Gender** | | | |
| Male | 27.43 (144) | | |
| Female | 66.29 (348) | | |
| Transgender and Nonbinary | 6.28 (33) | | |
| **Race** | | | |
| White | 87.62 (460) | | |
| Non-white | 12.38 (65) | | |
| **Education** | | | |
| First in family to college | 8.95 (47) | | |
| Not first in family to college | 91.05 (478) | | |
| **Outcome variables** | **Mean (SD)** | **Study Variance** | **Weekly Variance** |
| PSS | 15.85 (7.33) | 10.65 (5.03) | – |
| PSS Change between weeks | -0.48 (5.90) | 12.07 (7.88) | - |
| PSS Deviation from study mean | -1.15e-08 (4.11) | 10.65 (5.03) | - |
| **Time-varying independent predictors** | **Mean (SD)** | **Study Variance** | **Weekly Variance** |
| TST (hours), all days | 7.41 (0.83) | 5.78 (1.47) | 3.30 (1.48) |
| TST (hours), weekday | 7.44 (0.86) | 5.15 (1.49) | 2.72 (1.40) |
| Bedtime, all days | 12: 34 AM (75.47 min) | 409.73 (195.89) | 215.79 (126.80) |
| Bedtime, weekday | 12: 26 AM (77.12 min) | 354.93 (176.37) | 171.03 (108.71) |
| Waketime, all days | 9:03 AM (72.74 min) | 384.56 (202.53) | 192.32 (119.76) |
| Waketime, weekday | 8:56 AM (77.12 min) | 335.71 (184.03) | 152.62 (108.52) |
| Deep sleep (%) | 32.18 (9.63) | 32.35 (8.84) | 17.36 (7.33) |
| REM sleep (%) | 20.39 (7.07) | 21.90 (7.61) | 11.05 (5.10) |
| Light sleep (%) | 47.43% (8.71) | 31.79 (7.78) | 17.44 (7.41) |
| Efficiency (%) | 87.13 (4.46) | 19.69 (7.29) | 9.79 (5.87) |
| Wake up count (times) | 8.54 (2.36) | 10.90 (2.47) | 5.89 (2.54) |
| SOL (min) | 11.79 (6.61) | 37.22 (23.96) | 18.21 (15.50) |
| RHR (bpm) | 62.75 (8.64) | 18.90 (8.38) | 7.89 (5.62) |
| HRV (ms) | 68.24 (33.09) | 57.95 (32.36) | 26.84 (20.64) |
| ARR (breaths per min) | 15.56 (1.61) | 2.42 (1.03) | 1.02 (0.65) |
| Skin temperate deviation (degrees C) | 0.14 (0.20) | 0.79 (0.50) | 0.26 (0.25) |

binary outcome (Tables 2 and 3). In modeling PSS as a continuous outcome, TST, RHR, HRV, and ARR were significant predictors), controlling for gender and week of the semester (Table 2). In model 1b, an additional hour of sleep was associated with a 0.877 decrease in PSS ($p<0.01$). In model 1c, an increase of 1 bpm in RHR was associated with a 0.055 increase in PSS ($p<0.01$). In model 1d, an increase of 1 ms in HRV was associated with a 0.012 decrease in PSS ($p<0.05$). In model 1e, an increase of 1 breath per minute in ARR was associated with a 0.270 increase in PSS ($p<0.05$). In addition to being significant predictors of PSS, the addition of TST, RHR, HRV, and ARR improved the model fit. The addition of TST having the greatest improvement in model fit (Table 2).

We also present the results from models with PSS as a binary outcome, moderate-to-high stress (PSS> = 14), due to its clinical relevance. In modeling stress as a binary outcome, TST, RHR, HRV, and ARR were significant predictors of participants reporting symptoms of moderate-to-high stress, controlling for gender and week of the semester (Table 3). For every additional hour of sleep (model 2b), the odds of reporting moderate-to-high symptoms of stress decreased by 0.617 ($p<0.01$). For every 1 bpm increase in RHR (model 2c), the odds of reporting symptoms of moderate-to-high stress increased by 1.036 ($p<0.05$). For every 1 ms increase

**Table 2. Model fit for Mixed-Effects Multi-linear Regressions with PSS as a continuous outcome measure.** In this model the fixed effects are gender, week, and sleep measures TST, RHR, HRV, and ARR. The random effects are participant and week of study. The likelihood ratio (LR) compares the model to the previous model fit.

| | Predictor | Coefficient (SE) | p-value | LR chi2 | LR p-value | AIC | BIC |
|---|---|---|---|---|---|---|---|
| 1 | Week | -0.346 (0.047) | 0.000 | | | 19307.07 | 19337.29 |
| 1a | Week | -0.347 (0.046) | 0.000 | | | | |
| 1a | Gender (nonmale) | 2.956 (0.571) | 0.000 | 26.09 | 0.0000 | 19282.98 | 19319.24 |
| 1b | Week | -0.331 (0.046) | 0.000 | | | | |
| 1b | Gender (nonmale) | 3.357 (0.573) | 0.000 | | | | |
| 1b | Mean TST (hours) | -0.877 (0.138) | 0.000 | 40.15 | 0.0000 | 19244.83 | 19287.13 |
| 1c | Week | -0.357 (0.047) | 0.000 | | | | |
| 1c | Gender (nonmale) | 2.543 (0.591) | 0.000 | | | | |
| 1c | Mean RHR (bpm) | 0.055 (0.021) | 0.009 | 6.78 | 0.0092 | 19278.21 | 19320.51 |
| 1d | Week | -0.355 (0.046) | 0.000 | | | | |
| 1d | Gender (nonmale) | 2.814 (0.574) | 0.000 | | | | |
| 1d | Mean HRV (ms) | -0.012 (0.006) | 0.035 | 4.45 | 0.0348 | 19280.53 | 19322.83 |
| 1e | Week | -0.343 (0.047) | 0.000 | | | | |
| 1e | Gender (nonmale) | 2.762 (0.577) | 0.000 | | | | |
| 1e | Mean ARR (breaths/min) | 0.270 (0.131) | 0.040 | 4.23 | 0.0397 | 19280.75 | 19323.06 |

in HRV (model 2d), the odds of reporting symptoms of moderate-to-high of stressed decreased by 0.988 ($p<0.05$). For each additional breath per minute (model 2e), the odds of reporting symptoms of moderate-to-high stress increased by 1.230 ($p<0.01$).

Lastly, we investigated the association between changes in stress with changes in sleep estimates (Table 4). There was a significant relationship between average TST and change in participant PSS between weeks of the study ($p<0.01$). There was also a significant relationship between deviation in TST from a participant's study average and change in PSS between weeks ($p<0.01$), deviation in RHR from a participant's study average and change in PSS ($p<0.01$), and deviation in HRV from a participant's study average and change in PSS ($p<0.01$).

**Table 3. Model fit for Mixed-Effects Multi-linear Regressions with PSS-Moderate (PSS > = 14) as a binary outcome measure.** In this model the fixed effects are gender, week, and mean sleep measures TST, RHR, HRV, and ARR. The random effects are participant and week of study. The likelihood ratio (LR) compares the model to the previous model fit.

| | Predictor | Coefficient (SE) | OR | p-value | LR chi2 | LR p-value | AIC | BIC |
|---|---|---|---|---|---|---|---|---|
| 2 | Week | -0.111 (0.034) | 0.895 | 0.001 | | | 3071.054 | 3095.226 |
| 2a | Week | -0.117 (0.033) | 0.890 | 0.000 | | | | |
| 2a | Gender (nonmale) | 1.597 (0.314) | 4.953 | 0.000 | 26.56 | 0.0000 | 3046.493 | 3076.708 |
| 2b | Week | -0.109 (0.033) | 0.897 | 0.001 | | | | |
| 2b | Gender (nonmale) | 1.853 (0.324) | 6.379 | 0.000 | | | | |
| 2b | Mean TST (hours) | -0.483 (0.100) | 0.617 | 0.000 | 24.22 | 0.0000 | 3024.273 | 3060.531 |
| 2c | Week | -0.123 (0.033) | 0.884 | 0.000 | | | | |
| 2c | Gender (nonmale) | 1.338 (0.328) | 3.811 | 0.000 | | | | |
| 2c | Mean RHR (bpm) | 0.035 (0.014) | 1.036 | 0.010 | 6.67 | 0.0098 | 3041.818 | 3078.076 |
| 2d | Week | -0.126 (0.033) | 0.882 | 0.000 | | | | |
| 2d | Gender (nonmale) | 1.475 (0.316) | 4.371 | 0.000 | | | | |
| 2d | Mean HRV (ms) | -0.012 (0.004) | 0.988 | 0.035 | 11.54 | 0.0007 | 3036.95 | 3073.208 |
| 2e | Week | -0.114 (0.033) | 0.892 | 0.001 | | | | |
| 2e | Gender (nonmale) | 1.452 (0.317) | 4.272 | 0.000 | | | | |
| 2e | Mean ARR (breaths/min) | 0.207 (0.080) | 1.230 | 0.010 | 6.85 | 0.0089 | 3041.641 | 3077.899 |

**Table 4. Association between sleep measures and changes in stress and deviations in stress among first-year college students (N = 3,112).** In this model the fixed effects are gender, week, and the sleep measures TST, RHR, HRV, and ARR. The random effects are participant and week of study. The likelihood ratio (LR) compares the model to the previous model fit.

| | Δ PSS | | | | σ PSS | | | |
|---|---|---|---|---|---|---|---|---|
| | Coef. | SE. | 95% CI | Sig. | Coef. | SE. | 95% CI | Sig. |
| TST (Hrs) | -0.584 | 0.151 | [-0.881, 0.288] | **0.000** | -0.386 | -0.089 | [-0.560, -0.212] | **0.000** |
| RHR (bpm) | 0.017 | 0.015 | [-0.011, 0.046] | 0.237 | 0.003 | 0.009 | [-0.014, 0.020] | 0.769 |
| HRV (ms) | -0.005 | 0.004 | [-0.013, 0.003] | 0.230 | 0.000 | 0.002 | [-0.005, 0.004] | 0.906 |
| ARR (breaths per min) | -0.039 | 0.080 | [-0.196, 0.118] | 0.624 | 0.026 | 0.047 | [-0.066, 0.117] | 0.583 |
| **Deviation in Values (Weekly Measure–Participant Mean)** | | | | | | | | |
| TST (Hrs) | -1.312 | 0.227 | [-1.756, 0.868] | **0.000** | -0.900 | 0.135 | [-1.165, -0.635] | **0.000** |
| RHR (bpm) | 0.188 | 0.041 | [0.107, 0.269] | **0.000** | 0.020 | 0.024 | [-0.028, 0.068] | 0.410 |
| HRV (ms) | -0.048 | 0.013 | [-0.073, -0.023] | **0.000** | -0.003 | 0.008 | [-0.018, 0.012] | 0.700 |
| ARR (breaths per min) | 0.180 | 0.339 | [-0.484, 0.845] | 0.595 | 0.423 | 0.189 | [0.052, 0.794] | 0.025 |
| **Weekly Variance (Participant Max–Participant Min)** | | | | | | | | |
| TST (Hrs) | 0.039 | 0.085 | [-0.127, 0.205] | 0.647 | -0.035 | 0.050 | [-0.133, 0.063] | 0.488 |
| RHR (bpm) | 0.012 | 0.019 | [-0.026, 0.050] | 0.542 | 0.007 | 0.011 | [-0.016, 0.029] | 0.563 |
| HRV (ms) | -0.002 | 0.005 | [-0.012, 0.009] | 0.771 | 0.001 | 0.003 | [-0.005, 0.007] | 0.744 |
| ARR (breaths per min) | -0.119 | 0.173 | [-0.459, 0.221] | 0.491 | 0.171 | 0.099 | [-0.023, 0.365] | 0.084 |

Additionally, there was a significant relationship between TST and deviation in PSS from a participant's study mean ($p<0.01$) as well as between deviation in TST from a participant's study average and deviation in PSS from a participant's study mean ($p<0.01$). There were no significant associations between weekly variance in sleep measures and change in PSS nor between weekly variance in sleep measures and deviation in PSS.

To check the robustness of our results from mixed-effects models, we also explored mixed effects models that use gradient-boosted trees to assess dominant predictors of stress in our sample (S2 Text). Results were consistent with the results of our mixed-effects multilevel regression models (S8 Table).

## Discussion

Young adults are at risk for concerning mental health symptoms [50,51], especially those undergoing major life transitions like the start of college [52]. Mounting evidence has shown that the mental health of college students was severely affected by the COVID-19 pandemic [53–55], and that the rise in mental health burden has remained high in this population [56]. There has been much discussion about how to address the growing need for mental health support, particularly on college campuses [57]. Researchers and health professionals have long suspected that stress plays a role in sleep, yet prior work has primarily relied upon cross-sectional, retrospective, and self-report assessments of sleep to assess the relationship between these measures. Subjective accounts of sleep duration are not reliable or objective measures of sleep duration; single time point assessment is also vulnerable to bias and has validity concerns.

Wearable devices estimating sleep have been suggested as a potential mechanism for identifying changes in mental health status in college students and prompting interventions [30] because they provide a more consistent picture of vital sign measures that are linked to health [58]. Recent work has shown that sleep abnormalities may be useful for predicting changes in mental health [29], however, many epidemiologic studies using biometric wearables have been limited by small sample sizes or short durations. To address these limitations, we enrolled a

large cohort of college students and collected surveys and continuous biometric monitoring during the first semester of college. In our cohort, 64% of perceived stress scores were above the threshold for moderate-to-high stress. The average TST for the survey period was 7.41 hours ($SD$ = 0.83 hours); however, the average minimum night of sleep for our cohort was 5.75 hours ($SD$ = 1.14 hours) which is well below the recommended guidelines for young adults. Furthermore, 32% of our sample slept, on average, less than 7 hours. While these distributions of sleep may seem insufficient, they are higher than other comparable collegiate samples pointing to the widespread disrupted sleep for undergraduate students.

We show that estimates of sleep measures from a consumer wearable in first-year college students were significantly associated with weekly PSS, a marker of mental health and well-being. Decreased TST, increased RHR, decreased HRV, and increased ARR were significantly associated with higher PSS and the likelihood of reporting symptoms consistent with a moderate-to-high level of stress (Tables 2 and 3). Gender explained some of the variance in stress measures. As expected from previous studies [59,60], participants who did not identify as male (i.e., female, nonbinary, and transgender students) had significantly higher levels of perceived stress than male participants (S2 Table). The inclusion of sleep measures as predictors in mixed-effects models significantly improved model fit and the explanatory value of our models (Tables 2 and 3). In our models, TST had the most substantial improvement in model fit measured by model fit estimates (i.e., AIC, BIC, and likelihood ratio) (Tables 2 and 3). These associations provide evidence that there is a relationship between changes in stress and changes in sleep measures, particularly TST (Table 4). However, the practical significance of these relationships is important to note.

Our study adds to the growing literature utilizing sleep and cardiorespiratory markers to identify changes in health [36,61,62]. Mechanistically there is a relationship between stress and decreased parasympathetic regulation [63] which modulates the neural pathways affecting heart rate, heart rate variability, and respiratory rate [64]. When an individual is under physical or mental stress, parasympathetic activity decreases and sympathetic activity increases resulting in increased RHR and decreased HRV. HRV has been linked to sleep stages and higher night-time HRV has been linked to better sleep quality [63,65]. Importantly, cardiovascular mortality has been associated with low HRV [43]. Changes in these cardiorespiratory measures have been detected in college students following stressful exposures [36].

The inclusion of ARR as a predictor of stress in our study is unique in the literature, where a growing number of studies have pointed to TST, HRV, RHR, and sleep stages as sleep measures that are associated with stress. Stress has also been associated with increased respiratory rate and respiratory variability [66]. ARR has been referred to as a neglected vital sign due to its importance in health assessment, but relatively minimal focus in the literature [67]. The stability of nightly ARR within healthy individuals, particularly the low internight variability in ARR [68], makes it a potentially useful metric for tracking changes in wellness and has been used to assess stress reactivity [69–71]. ARR has gained attention due to its relationship to COVID-19 infection [68] and increases in 3–5 breaths/min can signal health deterioration [72]. However, the low variance of this measure and the potential for measurement error exists in our data. In our analysis 93% of the variation in ARR was due to differences between individuals limiting its (S7 Table). A large study validating the respiratory estimates from Oura with PSG is needed to further understand this potentially useful measure.

Longer term studies with further investigation of temporal correspondence between stress and sleep measure deviations may provide better understanding of causal relationships as well as how individual factors affect these relationships. Further analysis of the onset of stress, its duration, and the temporal relationship to sleep disturbances may provide greater clarity on whether sleep measures associated with stress can be used to predict stress in this population.

During exploratory analyses, we detected significant differences in sleep and stress measures during the Thanksgiving break. Utilizing school breaks as a quasi-experimental study for the influence of reduced academic stressors on sleep is a worthwhile area for future study. To our knowledge, there is no published evidence of large, randomized control trials of long-term behavioral interventions targeting stress in college students with continuous wearable data. These studies could provide more insight into whether reductions in stress are associated with changes in sleep measures over longer time periods. Future research could also evaluate the potential impact of mobile app-based interventions following the detection of sleep disturbances.

There are limitations to the present study. The current study focused on raw sleep measures that could be extracted from the Oura ring's longest nightly sleep period. It will be important for future studies to evaluate additional sleep variables, such as daytime naps, which have been associated with mental health in college students [30]. While sleep data was taken nightly, surveys were collected on a weekly basis; aggregation of an individual's sleep measures by week could have introduced bias. Second, we do not have stress or sleep data before participants moved to college and therefore cannot assess the influence of beginning college on stress and sleep measures and how these health behaviors may have shifted at the start of college. Expanding this window of analysis may identify whether the link between sleep measures and PSS holds prior to the beginning of college and throughout the college experience. Third, our sample was predominantly female, white, and were not first-generation college students (Table 1). Future work would benefit from inclusion of additional cohorts to assess the influence of these traits and applicability of these results to a broader population. Compared to previous studies on first-year college students [11–13], this cohort got more sleep pointing to potential differences in this population from other college-aged groups. Fourth, potential confounders also exist in our data. In our analysis, we did not account for some factors that may influence sleep measures such as psychotropic medications, sleep disorders, physical activity, or the use of substances (e.g., caffeine, marijuana, and alcohol use).

The first year of college is a particularly important period for understanding sleep patterns and stress. As students transition to a time during which they have more autonomy, they establish sleep habits in the context of increased academic pressure, changes in their social milieu, and the development of adult coping behaviors. The present work highlights the potential utility of monitoring sleep, suggesting that these measures may identify within individual changes that are concerning for stress. As the demand for mental health services grows, determining which wearable-derived sleep estimates provide information about well-being and can predict worsening mental health in young adults is an important area of study.

## Conclusions

In their first semester of college, students are experiencing a big life change that has been associated with both sleep dysregulation and increased stress. The present work provides support for the use of sleep estimates from wearable devices in the prediction of perceived stress in first-year college students. Over 500 first-year students had an average of 7 hours and 26 minutes of sleep per night and showed elevated perceived stress scores over the course of their fall semester of college. The major strength of the present work is that after considering multiple factors that are well-known to influence stress in young adults, there was a persistent significant relationship between perceived stress and indices of physiologic stress–reduced sleep duration, increased resting heart rate, decreased night-time heart rate variability, and increased night-time average respiratory rate. Findings from the Oura ring suggest that decreased sleep, increased resting heart rate, decreased night-time heart rate variability, and

increased night-time respiratory rate are predictive of increased perceived stress and the likelihood of experiencing symptoms consistent with moderate-to-high levels of stress, measured at weekly intervals after accounting for gender and week of the semester. The present findings call for more research on the utility of wearable data to identify which young adults are at greatest risk for elevated stress given the implications for increased morbidity and mortality associated with mental health for this population.

## Supporting information

**S1 Text. Stress Assessment.**
(DOCX)

**S2 Text. Additional Methods: Nonlinear Models.**
(DOCX)

**S1 Data. Sleep Measures Data.**
(CSV)

**S1 Fig. Participants with survey and Oura data.**
(DOCX)

**S1 Table. Oura Ring measures, descriptions and units.**
(DOCX)

**S2 Table. Distributions of outcome variables and sleep measures by dataset inclusion and exclusion criteria for participants with at least 3 nights of Oura data per week.**
(DOCX)

**S3 Table. Distributions of outcome variables and sleep measures by dataset inclusion and exclusion criteria for participants with at least 1 night of Oura data per week.**
(DOCX)

**S4 Table. Correlation between predictor variables and PSS.**
(DOCX)

**S5 Table. Single variable sleep mixed-effects MLR models (N = 3,112) with 525 participants.** Model $Y_{ij} = \beta_0 + \beta_1$ X sleep measure$ij + \beta_2$ X weeknum$_{ij} + u_j + \varepsilon_{ij.}$
(DOCX)

**S6 Table. Demographic variables as predictors of PSS.**
(DOCX)

**S7 Table. Proportion of variance in outcome and predictor variables attributable to differences between subjects.**
(DOCX)

**S8 Table. Nonlinear predictive models in GPBoost for PSS outcome.**
(DOCX)

## Acknowledgments

We want to thank MassMutual for supporting our research focused on the health and well-being of college students. This work was made possible by MassMutual under Grant Number FP 2860.

## Author Contributions

**Conceptualization:** Laura S. P. Bloomfield, Ellen W. McGinnis, Ryan S. McGinnis, Matt Price, Taylor H. Ricketts, Peter Sheridan Dodds, Christopher M. Danforth.

**Data curation:** Laura S. P. Bloomfield, Mikaela I. Fudolig.

**Formal analysis:** Laura S. P. Bloomfield, Mikaela I. Fudolig.

**Funding acquisition:** Laura S. P. Bloomfield, Juniper L. Lovato, Peter Sheridan Dodds, Christopher M. Danforth.

**Investigation:** Laura S. P. Bloomfield.

**Methodology:** Laura S. P. Bloomfield, Mikaela I. Fudolig.

**Project administration:** Laura S. P. Bloomfield, Julia Kim, Jordan Llorin, Kathryn Stanton.

**Supervision:** Christopher M. Danforth.

**Visualization:** Laura S. P. Bloomfield.

**Writing – original draft:** Laura S. P. Bloomfield.

**Writing – review & editing:** Laura S. P. Bloomfield, Mikaela I. Fudolig, Juniper L. Lovato, Ellen W. McGinnis, Ryan S. McGinnis, Matt Price, Taylor H. Ricketts, Peter Sheridan Dodds, Christopher M. Danforth.

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
