## [Decision Letter · Decision Letter 0]

16 Oct 2023

PDIG-D-23-00273

Detecting stress in college freshman from wearable sleep data

PLOS Digital Health

Dear Dr. Bloomfield,

Thank you for submitting your manuscript to PLOS Digital Health. After careful consideration, we feel that it has merit but does not fully meet PLOS Digital Health's publication criteria as it currently stands. Therefore, we invite you to submit a revised version of the manuscript that addresses the points raised during the review process.

Please submit your revised manuscript within 60 days Dec 15 2023 11:59PM. If you will need more time than this to complete your revisions, please reply to this message or contact the journal office at digitalhealth@plos.org. Please include the following items when submitting your revised manuscript:

We look forward to receiving your revised manuscript.

Kind regards,

Raquel Simões de Almeida, PhD

Academic Editor

PLOS Digital Health

Journal Requirements:

2. We ask that a manuscript source file is provided at Revision. Please upload your manuscript file as a .doc, .docx, .rtf or .tex.

3. Please provide separate figure files in .tif or .eps format only and remove any figures embedded in your manuscript file. Please also ensure that all files are under our size limit of 10MB.

4. We have noticed that you have uploaded Supporting Information files, but you have not included a list of legends. Please add a full list of legends for your Supporting Information files after the references list.

5. In the online submission form, you indicated that "This is an ongoing study and therefore not all data will be available until the study conclusion to protect enrolled participants. Requests for data sharing and requests accessibility should be directed to the corresponding author". All PLOS journals now require all data underlying the findings described in their manuscript to be freely available to other researchers, either 1. In a public repository, 2. Within the manuscript itself, or 3. Uploaded as supplementary information.

Additional Editor Comments (if provided):

The paper "Detecting stress in college freshman from wearable sleep data" presents an interesting and timely topic in the field of stress detection and monitoring. The literature review is well-structured and provides a solid foundation for the research. However, it is crucial to address certain aspects of the methodology and data analysis to enhance the rigor and clarity of the study. Firstly, they should address the unexpected association between perceived stress and autonomic markers, particularly the use of ARR instead of HR/HRV, providing explanations and considering potential measurement errors. Secondly, the authors should explore within-subject effects (if it is possible) to investigate how fluctuations in stress ratings over time affect autonomic markers, ensuring that the study capitalizes on its design for a more comprehensive analysis. Additionally, compliance and data collection details should be presented more prominently to establish the reliability of wearable device data. Furthermore, it's essential to differentiate between weekend and weekday sleep, given the distinctive sleep patterns among college students. Effect size, as well as statistical significance, should be calculated and presented to convey practical significance, addressing the wide variance in ARR relative to PSS. Other points made by the reviewers should be addressed. By attending to these key points, the paper will be better positioned to contribute significantly to the field.

Reviewers' comments:

Reviewer's Responses to Questions

**Comments to the Author**

1. Does this manuscript meet PLOS Digital Health’s publication criteria? Is the manuscript technically sound, and do the data support the conclusions? The manuscript must describe methodologically and ethically rigorous research with conclusions that are appropriately drawn based on the data presented.

Reviewer #1: Partly

Reviewer #2: Partly

Reviewer #3: Yes

2. Has the statistical analysis been performed appropriately and rigorously?

Reviewer #1: I don't know

Reviewer #2: N/A

Reviewer #3: Yes

3. Have the authors made all data underlying the findings in their manuscript fully available (please refer to the Data Availability Statement at the start of the manuscript PDF file)?

Reviewer #1: No

Reviewer #2: No

Reviewer #3: Yes

4. Is the manuscript presented in an intelligible fashion and written in standard English?

Reviewer #1: Yes

Reviewer #2: Yes

Reviewer #3: Yes

5. Review Comments to the Author

Reviewer #1: A revealing study, current and relevant to the field of mental health and sleep studies, using a recent artificial intelligence device. However, there are still doubts about the prior validation by an ethics committee, as the opinion or registration number of the process submitted for evaluation was not provided in the annex, and, on the other hand, how the anonymity of the participants was guaranteed (procedures for storing, processing and accessing health/personal data were not specified). On the other hand, nowhere in the article does it clarify the clinical validation of the OURA ring and its comparability with the PSG (the study identified as a reference for this validation was carried out on a small sample, in a different age group, especially in terms of behavior). The study did not include exclusion criteria, which are mentioned here as one of the limitations. Instead of presenting this limitation, it should have been considered as a separate sub-group, or effectively eliminated in order to see if there were any changes to the results obtained. Even in terms of psychiatric pathology, should be considered as an exclusion factor or analyzed as an independent group, given the incidence of some pathologies in the adolescence-adulthood transition.

Reviewer #2: The authors report a large-scale observational study using Oura rings to measure sleep and nocturnal physiology, and self-report measures of perceived stress and mental health history to investigate the association between the objective measures of sleep and physyiology and the subjective psychological measures. The findings show that sex, mental health history and average respiration rate (ARR) are significantly associated with perceived stress.

First, the authors must be commended for conducting a study such as this. Collecting data at this scale and over an extended period is no mean feat. 

The report is concise and to the point; however, there are several issues that need to be addressed. From a conceptual viewpoint items #3,

Design and analysis

1. I thought the literature review on prior work on autonomic markers of stress was excellent, as were references comparing the sensitivity of HR and respiration measures to different experimental stress tests. Given the knowledge that stress through sympathetic activation increases both HR, decreases HRV and increases respiratory rate, It is of major concern that the hypothesized association between stress (PSS) and HR/HRV was not found, but instead, an association was with ARR alone. We don't know the MAE of ARR measurement with Oura (we do know the errors relative to PSG for sleep and HR/HRV with ECG). Published data that the authors cite show that the MAE from PPG derived Respiration Rate is 2-3 breaths/min which is within the noise levels of the effect documented here. There needs to be some explanation for this especially when the latter is inferred from the PPG signal yields HRV data as well. 

2. The value of a mixed effects model is often revealed in within-subject effects, in this case, instances where higher tri-weekly stress ratings would affect autonomic markers. Identifying how greater self-ratings of stress relate to differences in autonomic markers of stress would surely be interesting. The authors have the design to investigate fluctuations of stress over time. If this is what the authors are already showing, they should be explicit about it. If not, perhaps they could consider adding an analysis to test whether changes in PSS affect markers of interest. 

3. In a longitudinal study like this, a clear indication of compliance / acceptance needs to be stated in the results and not buried in Pg. 19 of supplementary material. Overall, the wear rate was quite good. It would also be good to know at the outset how many weeks (7) data was collected for. 

4. Regarding periods of non-wear of the rings. Data was included (and averaged across the week) if at least 3 days of data was present. Our experience with missing data is that it might not be an issue given the overall compliance here. However, this may be an issue if the missingness of the data is not random (e.g., if a participant removes the ring when they are studying late and forgets to put it back on). More data showing the extent and distribution of the missing data, and possibly analysis to show it is not a concern would help the reader have more confidence in the data. 

5. Given that the sleep durations are quite good for college students, it would be helpful to differentiate between weekend and weekday sleep, which we know differ in timing and duration in this population. Including this in the analysis may provide more insight into sleep patterns and how they relate to stress (given their long TST, it may be that these students are primarily wearing the rings at the weekends).

6. With large datasets, it is important to convey the effect size in addition to statistical significance as a significant factor that has a tiny effect size may not be practically meaningful. This should be calculated and displayed in the tables. Of concern is the scatter plot in Fig 1 shows the wide variance in ARR relative to PSS.

7. As with point #3 it would be helpful if the model(s) was specified in the main text. Table 1 is not interpretable by itself without that information. 

8. It would be helpful to see a table of the correlations between the various independent variables, to get an idea of the structure of the data. Higher ARR is probably correlated with being female, so that might explain the relationship (while that should be accounted for in the model, but with high collinearity, there might still be some effect). Knowing how the independent variables relate to each other would give the reader some idea about this. 

Other issues

In Table 1, and throughout, should probably be referred to as Heart Rate Variability. Also, in the lower half of Table 1, there is no indication what the values represent. I assume it is PSS score, but this should be stated clearly.

When comparing models, predictors are usually added to models, which are then tested against the simpler model to see if they add explanatory power. A slightly different approach is taken in this paper, which isn’t necessarily wrong; however, we don’t know whether adding mental health history to the demographic factors significantly improves the model fit.

The first sentence of the abstract is not correct. Recent works by Yap (Ann Behav Med 2022), Menghini (Sleep Health 2023), Ng (Sleep Medicine 2023) have sought to relate sleep to mental-health markers/ stress. Also, in the abstract, the language describing the relationship between ARR, and stress seems to suggest that higher ARR results in stressful experiences, when the relationship is probably the other way round.

Wearables do not measure contributors to sleep disturbances or mental health – all they do is provide estimates of sleep (and physiological) measures. Discovering what contributes to sleep disturbance and mental health issues requires significantly more. 

The title is similarly imprecise, the study does not aim to detect stress, rather it attempts to predict perceived stress. Furthermore, the data collected in this study is not wearable – it is collected using wearable devices.

Reviewer #3: this is well written paper in important topic

I am interested in the internal consistently and intraclass correlation of the data structure

discussion need to be developed further 

references need attention for their format intext

6. PLOS authors have the option to publish the peer review history of their article (what does this mean?). If published, this will include your full peer review and any attached files.

**Do you want your identity to be public for this peer review?** For information about this choice, including consent withdrawal, please see our Privacy Policy.

Reviewer #1: No

Reviewer #2: No

Reviewer #3: Yes: none

---

## [Editor Report · Decision Letter 1]

9 Jan 2024

PDIG-D-23-00273R1

Predicting stress in college freshmen using sleep data from wearable devices

PLOS Digital Health

Dear Dr. Bloomfield,

Thank you for submitting your manuscript to PLOS Digital Health. After careful consideration, we feel that it has merit but does not fully meet PLOS Digital Health's publication criteria as it currently stands. Therefore, we invite you to submit a revised version of the manuscript that addresses the points raised during the review process.

Please submit your revised manuscript within 30 days Feb 08 2024 11:59PM. If you will need more time than this to complete your revisions, please reply to this message or contact the journal office at digitalhealth@plos.org. Please include the following items when submitting your revised manuscript:

We look forward to receiving your revised manuscript.

Kind regards,

Raquel Simões de Almeida, PhD

Academic Editor

PLOS Digital Health

Journal Requirements:

Additional Editor Comments (if provided):

I have had the opportunity to thoroughly review the revised manuscript titled "Predicting stress in college freshmen using sleep data from wearable devices" and I am pleased to report that the authors have diligently addressed all the comments provided by the reviewers, and the overall quality of the paper has significantly improved as a result.

Nevertheless, the authors placed the "Materials and Methods" section after the "Conclusions", something that I don't think is very appropriate and that doesn't happen in other articles published on PLOS Digital Health. However, and despite this change which I think is necessary, I am confident in recommending its acceptance for publication. The manuscript now meets the high standards of the journal, and its dissemination would undoubtedly contribute valuable insights to the relevant academic community.
---

## [Editor Report · Decision Letter 2]

16 Feb 2024

Predicting stress in college freshmen using sleep data from wearable devices

PDIG-D-23-00273R2

Dear Dr. Bloomfield,

We are pleased to inform you that your manuscript 'Predicting stress in college freshmen using sleep data from wearable devices' has been provisionally accepted for publication in PLOS Digital Health.

Best regards,

Raquel Simões de Almeida, PhD

Academic Editor

PLOS Digital Health